# Comparative Thermoelectric Properties of Polypropylene Composites Melt-Processed Using Pyrograf® III Carbon Nanofibers

Antonio J. Paleo [1,*], Beate Krause [2], Ana R. Mendes [3], Carlos J. Tavares [4], Maria F. Cerqueira [5,6], Enrique Muñoz [7] and Petra Pötschke [2]

1   2C2T-Centre for Textile Science and Technology, University of Minho, Campus de Azurém, 4800-058 Guimarães, Portugal
2   Leibniz-Institut für Polymerforschung Dresden e.V. (IPF), Hohe Str. 6, 01069 Dresden, Germany
3   SEMAT/UM-Materials Characterization Services, University of Minho, 4804-533 Guimarães, Portugal; a85605@alunos.uminho.pt
4   Physics Centre of Minho and Porto Universities (CF-UM-PT), University of Minho, 4804-533 Guimarães, Portugal; ctavares@fisica.uminho.pt
5   INL—International Iberian Nanotechnology Laboratory, Av. Mestre, Jose Veiga, 4715-330 Braga, Portugal
6   CFUM—Center of Physics of the University of Minho, Campus de Gualtar, 4710-057 Braga, Portugal
7   Facultad de Física, Pontificia Universidad Católica de Chile, Santiago 7820436, Chile
*   Correspondence: ajpaleovieito@2c2t.uminho.pt

**Abstract:** The electrical conductivity (σ) and Seebeck coefficient (S) at temperatures from 40 °C to 100 °C of melt-processed polypropylene (PP) composites filled with 5 wt.% of industrial-grade carbon nanofibers (CNFs) is investigated. Transmission Electron Microscopy (TEM) of the two Pyrograf® III CNFs (PR 19 LHT XT and PR 24 LHT XT), used in the fabrication of the PP/CNF composites (PP/CNF 19 and PP/CNF 24), reveals that CNFs PR 24 LHT XT show smaller diameters than CNFs PR 19 LHT XT. In addition, this grade (PR 24 LHT XT) presents higher levels of graphitization as deduced by Raman spectroscopy. Despite these structural differences, both Pyrograf® III grades present similar σ (T) and S (T) dependencies, whereby the S shows negative values (n-type character). However, the σ (T) and S (T) of their derivative PP/CNF19 and PP/CNF24 composites are not analogous. In particular, the PP/CNF24 composite shows higher σ at the same content of CNFs. Thus, with an additionally slightly more negative S value, the PP/CNF24 composites present a higher power factor (PF) and figure of merit (zT) than PP/CNF19 composites at 40 °C. Moreover, while the σ (T) and S (T) of CNFs PR 19 LHT XT clearly drive the σ (T) and S (T) of its corresponding PP/CNF19 composite, the S (T) of CNFs PR 24 LHT XT does not drive the S (T) observed in their corresponding PP/CNF24 composite. Thus, it is inferred in PP/CNF24 composites an unexpected electron donation (n-type doping) from the PP to the CNFs PR 24 LHT XT, which could be activated when PP/CNF24 composites are subjected to that increase in temperature from 40 °C to 100 °C. All these findings are supported by theoretical modeling of σ (T) and S (T) with the ultimate aim of understanding the role of this particular type of commercial CNFs on the thermoelectrical properties of their PP/CNF composites.

**Keywords:** polypropylene; carbon nanofibers; thermoelectric properties; n-type polymer composites; electrical modeling

## 1. Introduction

In recent times, the electrical properties of conductive polymer composites (CPCs) based on thermoplastic polymers combined with conductive materials have been largely investigated [1], since their understanding is the basis for their applications as sensors, and devices for energy storage and harvesting [2]. In particular, the research on thermo-electric (TE) materials is an area of growing concern since it is considered an important

counterpart to renewable alternative energies such as solar and wind power [3]. Basically, TE materials are those that present positive or negative Seebeck coefficients (S), depending on if the potential ($\Delta$V) created by the temperature gradient ($\Delta$T) existing between the ends of the TE material is dominated by positive (holes) or negative (electrons) charge carriers [4]. Hence, the figure of merit (zT), defined as $\frac{S^2\sigma}{k}$T, drives the conversion efficiency in TE materials, where S is the Seebeck coefficient (also called thermopower), $\sigma$ is the electrical conductivity, and k is the thermal conductivity [5]. When producing CPCs, among other polymer matrices, polypropylene (PP) is one of the most studied materials, as it has optimal mechanical and thermal characteristics that are ideal for numerous applications [6]. Carbon nanofibers (CNFs), along with other conducting carbon materials such as carbon black (CB), carbon nanotubes (CNTs), and graphene, represent an attractive option for CPCs given their large surface area, remarkable aspect ratio (AR), high strength and storage modulus, and excellent thermal and electrical properties [7,8]. Moreover, since the distribution, dispersion and orientation of the CNFs within the PP affect the conductivity and the percolation threshold of CPCs, the correct choice of the processing method is also significant [8]. In this respect, melt-processing is usually preferred over other methods, such as solution mixing, since it prevents the need for solvents and enables mass production [9]. In addition, this technique allows CNFs to achieve good dispersion and distribution within the PP matrix at moderate shear mixing conditions [10]. Furthermore, it is useful to find direct correlations between the properties of CNFs and the derivative CPCs in order to adequately substitute conventional conducting materials. For example, Guadagno et al. reported that in CPCs made with epoxy resin and different CNFs, the CNFs with the highest AR showed lower electrical percolation thresholds and higher electrical conductivities compared to those with lower AR [11]. In another study, Silva et al. simulated the effect of AR on the $\sigma$ of CNT-based polymer composites and concluded that the CPCs with the highest electrical conductivities are achieved when using the CNTs with higher AR values [12]. Zie-Min et al. studied the relation between the AR of CNTs and the electrical percolation threshold in multiwall carbon nanotube (MWCNT)/thermoplastic elastomer (TPE) composites fabricated by melt mixing. The authors conclude that the diameter and length of CNTs considered individually can better explain the relation found between percolation threshold and the MWCNT dimensions rather than their combined AR values [13]. It is in this context that the present work is carried out. Based on previous studies [14,15], 5 wt.% of two Pyrograf® III CNFs (PR 19 LHT XT and PR 24 LHT XT) melt-processed with PP under the same conditions are morphologically, structurally, and thermally investigated by means of Scanning Electron Microscopy (SEM), X-ray diffraction (XRD), Raman spectroscopy and Differential scanning calorimetry (DSC). Moreover, the effects of the CNFs on the $\sigma$ and the Seebeck coefficient of the CPCs at temperatures from 40 °C to 100 °C are analyzed by examining the $\sigma$ (T) and S (T) of the as-received CNFs and the PP/CNF composites. The $\sigma$ (T) results of both (CNFs and PP/CNF composites) are theoretically supported by using the 3D variable range hopping (VRH) model [16], while their S (T) is depicted by the theoretical model proposed for describing the nonlinear S (T) of doped MWCNT mats [17]. Ultimately, the results discussed here may be useful to evaluate the effect of two different grades of Pyrograf® III CNFs on the thermoelectric properties of their derivative melt-processed PP/CNF composites.

## 2. Materials and Methods

### 2.1. Materials and Their Processing

A polypropylene powder, Daplen™ EE002AE, was used as a polymer matrix. Carbon nanofibers synthesized by chemical vapor deposition (CVD), Pyrograf® III PR 19 LHT XT and PR 24 LHT XT, (ASI, Cedarville, OH, USA), with average diameters of 150 nm and 100 nm, respectively, and lengths higher than 100 μm, according to the producer's datasheet [18], were applied. Both types of CNFs are grown by CVD at 1100 °C, and after they are heat-treated at 1500 °C in an inert atmosphere. This converts any chemically vapor-deposited carbon present on the surface to a short-range ordered structure. More

details about Pyrograf® III products can be found in previous reports [19]. In terms of morphology, both CNF types present a dual wall structure surrounding the hollow tubular core as shown in Figure 1 [14,15].

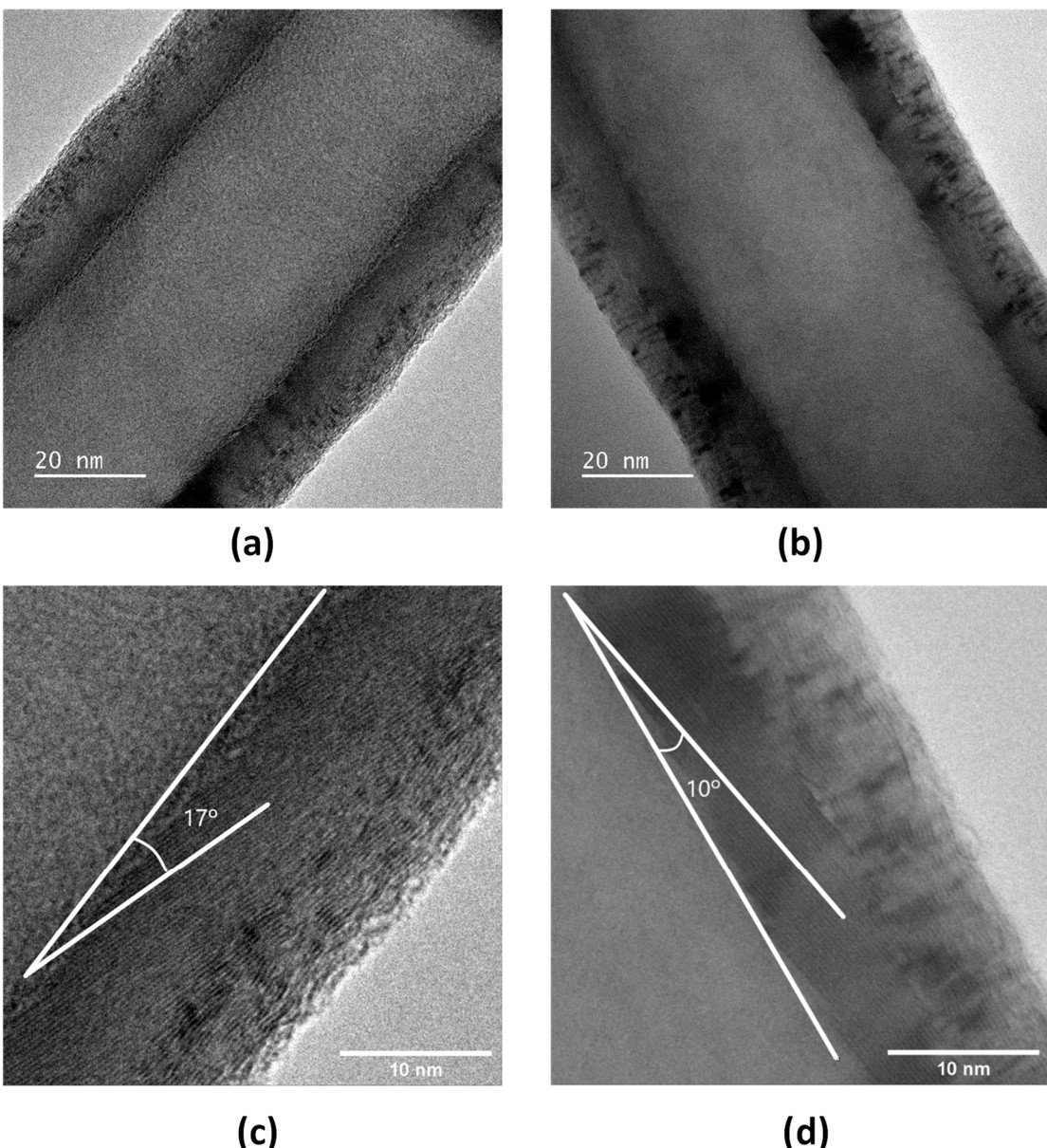

**Figure 1.** TEM images of Pyrograf® III carbon nanofibers: (**a**) PR 19 LHT XT and (**b**) PR 24 LHT XT. Detail of inner layers: (**c**) PR 19 LHT XT and (**d**) PR 24 LHT XT. ((**a**,**c**) reprinted with permission from Ref. [15]. (**b**,**d**) reprinted with permission from Ref. [14]).

Melt-mixed PP/CNF composites with 5 wt.% CNFs (above the electrical percolation threshold [20]) fabricated on a modular lab-scale intermeshing mini-co-rotating twin-screw extruder, with a screw diameter of 13 mm, barrel length of 338 mm and an approximate L/D ratio of 26, coupled to a cylindrical rod dye of approximate 2.85 mm diameter were studied. A detailed description of the melt extrusion conditions has been previously published [20]. The extruded PP/CNF composites were then pelletized and compression molded at 210 °C with a hot press PW40HT for 2 min (1.5 min pre-heating, ma. Force 50 kN, 0.5 min cooling in a minichiller, polyimide foil) to plates with a diameter of 60 mm and thickness of 0.5 mm. For thermal conductivity measurements, circular samples (diameter 12.5 mm, thickness 1.8 mm) were prepared under similar conditions. The codes used for PP/CNF composites

and their characteristics are summarized in Table 1. Photos of manufactured samples can be seen in the supporting information.

**Table 1.** CNFs and PP/CNF composites.

| PP/CNF Loading | CNF Type | Polypropylene | CNF Loading |
|---|---|---|---|
| PP/CNF19 | PR 19 LHT XT | Daplen$^{TM}$ EE002AE | 5 wt.% |
| PP/CNF24 | PR 24 LHT XT | | |

### 2.2. Morphological Analysis

The as-received CNFs were imaged with a transmission electron microscope (TEM, JEOL JEM-2100) utilizing a LaB6 electron gun at 80 kV and collected with a "OneView" 4k × 4k CCD camera at minimal under-focus to obtain the surface layers of the CNFs that were visible [14,15]. Morphological characterization of the PP/CNF composites was performed using scanning electron microscopy (SEM) by means of an Ultra plus microscope (Carl Zeiss GmbH, Germany, field emission cathode) at 3 kV. The compression molded plates as used for the TE measurements were cryo-fractured in liquid nitrogen and prior to observation, the surfaces were covered with 3 nm platinum.

### 2.3. XRD and Raman Analysis

The crystallographic structure of the as-received CNFs, PP, and the PP/CNF composites was investigated from X-ray diffraction (XRD) analysis, using a Bruker AXS D8 Discover diffractometer (Karlsruhe, Germany) operating at 40 kV and 40 mA in parallel geometry. Cu-K$\alpha$ was used as an X-ray source, with a wavelength of 1.54060 Å. All XRD patterns were acquired with a step size of 0.02° and an integration time of 2 s.

Raman spectroscopy measurements of PP/CNF24 were carried out on an ALPHA300 R Confocal Raman Microscope (WITec) using a 532 nm laser for excitation in backscattering geometry. The laser beam with $p$ = 0.5 mW was focused on the sample by a ×50 lens (Zeiss), and the spectra were collected with 600 groove/mm grating using five acquisitions with 2 s acquisition time. Raman spectroscopy of PP, CNFs, and PP/CNF19 presented here has been already published and obtained under the same conditions above described [15,21].

### 2.4. DSC Analysis

Differential scanning calorimetry tests (DSC) were operated in an argon atmosphere using a DSC Q20 instrument (TA Instruments, New Castle, DE, USA). In the non-isothermal experiments, the specimens were heated and cooled down at the rate of 10 °C min$^{-1}$, from 25 °C to 190 °C, and from 190 °C to 40 °C, respectively, to eliminate any previous thermal history. Following this preliminary step, the samples were heated up to 190 °C at 10 °C min$^{-1}$.

### 2.5. Thermoelectric Analysis

The Seebeck coefficient and electrical volume resistivity ($\rho$) of the PP/CNF composites and CNF powder were acquired using the self-constructed equipment TEG at Leibniz-IPF [22]. For the CNF powder, a PVDF tube (inner diameter 3.8 mm, length 16 mm) closed with copper plugs was utilized [23]. Photos and schema of the used thermoelectric measurement are detailed in supporting information (Figure S2). PP/CNF plates of 15 mm × 4.5 mm and painted at their ends with conductive silver ink were analyzed. The thermovoltage and electrical resistance were measured using the Keithley multimeter DMM2001 (Keithley Instruments, Cleveland, OH, USA) with a free insert length of 12 mm between the two copper electrodes. The volume resistivity was acquired using a 4-wire technique. The conductivity results represent the arithmetic means of ten measurements on two strips. The values of S were obtained at the mean temperatures of 40 °C, 60 °C, 80 °C, and 100 °C by implementing between the two copper electrodes gradient of temperatures up to ±8 K around that mean temperature in 2 K steps. The Seebeck coefficient was calcu-

lated as the average of eight thermoelectric voltage values at each temperature. This process was made three times, and the final means are reported. The thermal conductivity k of the PP/CNF composites was determined from the product of thermal diffusivity, density, and specific heat capacity. The thermal diffusivity was measured on disc samples (diameter 12.5 mm, thickness 1.8 mm) through the plate thickness using the light flash apparatus LFA 447 NanoFlash (Netzsch-Gerätebau GmbH, Selb, Germany) at 40 °C, 60 °C, 80 °C, and 100 °C. The specific heat capacity was calculated by comparing the signal heights between the PP/CNF composites and the reference Pyroceram 9606 (with known specific heat capacity) using the LFA 447 NanoFlash software. The density of the PP/CNF composites was determined using the buoyancy method. The given values represent the means of four measurements. The Seebeck coefficient, the electrical volume resistivity of CNFs, and PP/CNF19 presented here have been already published and obtained under the same conditions above described [15,21].

## 3. Results and Discussion

### 3.1. Morphological Analysis

Illustrative TEM images of Pyrograf® III CNFs PR 24 LHT XT and PR 19 LHT XT are shown in Figure 1 [14,15]. The total diameter of 25 individual CNFs was averaged from TEM analysis [20]. PR 19 LHT XT showed mean diameters of 110 nm, whereas the PR 24 LHT XT average was lower and around 80 nm. These diameters are similar to the sizes reported by Tessonnier et al. for the same CNFs [7] and lower than those given by the supplier [18]. The surfaces of both CNFs present a double structure, where the inner layers show parallel graphene sheets of different angles with respect to the hollow core (Figure 1c,d). The graphene sheets are also evident in the outer layers of both CNFs, though they are not stacked in parallel as in the inner layers. In addition, the contrast between the inner and outer layer structures is more evident in PR 24 LHT XT than in PR 19 LHT XT. Interestingly, it has been reported that PR 19 LHT XT has a CVD layer surrounding the nanofiber consisting of turbostatic carbon, whereas PR 24 LHT XT is synthesized at different conditions to eliminate that CVD layer [24], nonetheless Figure 1 shows that both CNFs are quite similar in terms of their outermost surface. The aspect ratios of both CNFs were estimated using the values of diameters above mentioned, and lengths provided by the supplier for PR 19 LHT XT (110 nm, 100 μm) and PR 24 LHT XT (80 nm, 100 μm). Hence, the aspect ratios of PR 19 LHT XT and PR 24 LHT XT are estimated as 909 and 1250, respectively. Therefore, the higher aspect ratio of PR 24 LHT XT should provide a lower electrical percolation threshold and better-developed networks due to a higher number of fibers at the studied content of 5 wt.% than in PP/CNF composites produced with PR 19 LHT XT having the lower AR. The SEM micrographs related to PP/CNF composites are shown in Figure 2. It is seen that the CNFs are nicely dispersed and homogeneously distributed. For both CNF types, fibers protrude from the polypropylene surface (Figure 2a,b), which is an indication of relatively low wettability and poor matrix-filler adhesion. About the same number and protruding lengths of CNFs are seen on the fracture surfaces. In an earlier study, it was shown in a detailed grayscale analysis of transmission light microscopy images on similar samples that the dispersion and distribution of PR 24 LHT XT in polypropylene are better than those of PR 19 LHT XT [20].

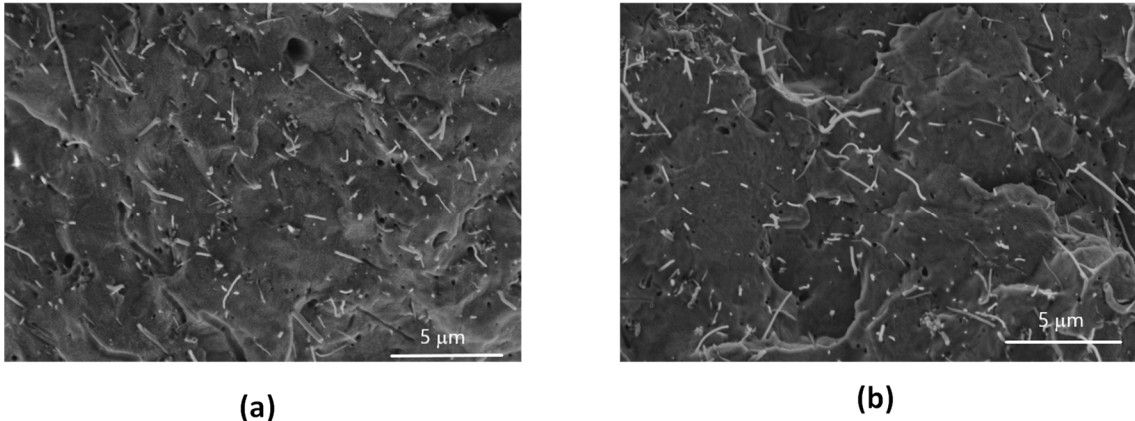

**Figure 2.** SEM micrographs of PP/CNF composites: (**a**) PP/CNF19 and (**b**) PP/CNF24.

*3.2. XRD and Raman Analysis*

Figure 3 exhibits the representative XRD patterns obtained from PP, as-received CNFs and PP/CNF composites in the 2θ range of 10°–50°. The strong diffraction peaks at approximately 14.08°, 16.86°, 18.58°, 21.20°, and 21.84° are assigned to the (110), (040), (130), (111), and (−131) crystal planes of α-phase PP crystallites, respectively, considering the ICDD 00-050-2397 crystallographic card. In the case of the CNF19 and CNF24 samples, a graphite phase (ICDD 01-071-4630 crystallographic card) was identified, alongside some minor intensity diffraction peaks and bands presented between 35° and 50°, which can be linked to impurities present in the nanofibers resulting from their manufacturing process. On the other hand, the XRD patterns of the PP/CNF composites show diffraction peaks of both α-phase PP and graphite phase, due to the inclusion of the CNFs in the PP matrix. However, the diffraction peak of the graphite phase is not clearly discerned as in the XRD patterns of both as-received CNFs. For the graphite phase, the diffraction peak intensity is related to its content in the composite. In this study, the content of CNFs is 5 wt.%; thus, the diffraction peak intensity of the graphite phase is much smaller when compared to the main diffraction peaks from the polymer.

A more detailed study of the diffraction patterns of the main reflections from polypropylene and from the PP/CNF composites is presented in the supporting information (Figure S3 and Table S1). All α-propylene phase diffraction peaks were individually fitted with Gaussian functions and their Bragg positions do not change significantly. It is interesting to see that the principal reflection from α-propylene (110) has its intensity decreased in the PP/CNF composites. Conversely, the (040) and (041) reflections are enhanced in the PP/CNF composites. The other reflections show minor variations. It is possible to follow the crystalline domain size variation of PP and the PP/CNF composites from Table S1. Generally, bulk PP has slightly larger crystalline domain sizes, varying between 14 and 20 nm. For the PP/CNF composites, the variation is narrower, between 14 and 18 nm, within the individual values slightly reduced for the PP/CNF24 composite, albeit within the experimental error (~10%).

A magnification of the diffraction patterns of Figure 3 in the region of the main graphitic contribution for the as-received CNFs and PP/CNF composites is also presented in the supporting information (Figure S4 and Table S2). All peaks were fitted with Gaussian functions and from this fit the respective line position contributions (2θ), the full width at half maximum (FWHM), and the crystalline domain size (D) were determined and presented in Table S2. The parameter D was determined from the Scherrer equation ($D = k\lambda/\beta\cos\theta$), in which k is the shape factor with an assumed value of 0.9 for these crystalline domains, λ is the X-ray wavelength, θ is the Bragg angle, and β is the FHWM of the respective XRD peak. It is apparent that the degree of crystallinity for the CNFs is higher than for the PP/CNFs. In the case of the CNFs, the main contributions from graphite arise from expanded (002) atomic planes crystallites in the out-of-plane direc-

tion (2θ = 25.8°–25.9°), while for the PP/CNF samples, these planes are more relaxed (2θ = 26.0°–26.2°), and closer to what is published in the graphite (ICDD 01-071-4630 crystallographic card). Interestingly, two graphitic crystalline domain sizes of 2 and 7 nm are clearly identified for both CNFs, which could be related to the two outer layers observed in TEM images (Figure 1). In addition, the graphitic crystalline domain sizes for the composites PP/CNF19 and PP/CNF 24 are larger, in the range of 9–14 nm. The diffraction peaks labeled as Peak 3 and Peak 4 (Figure S4 and Table S2) are ascribed to the (131) and (150) planes of PP (ICDD 00-061-1416 crystallographic card).

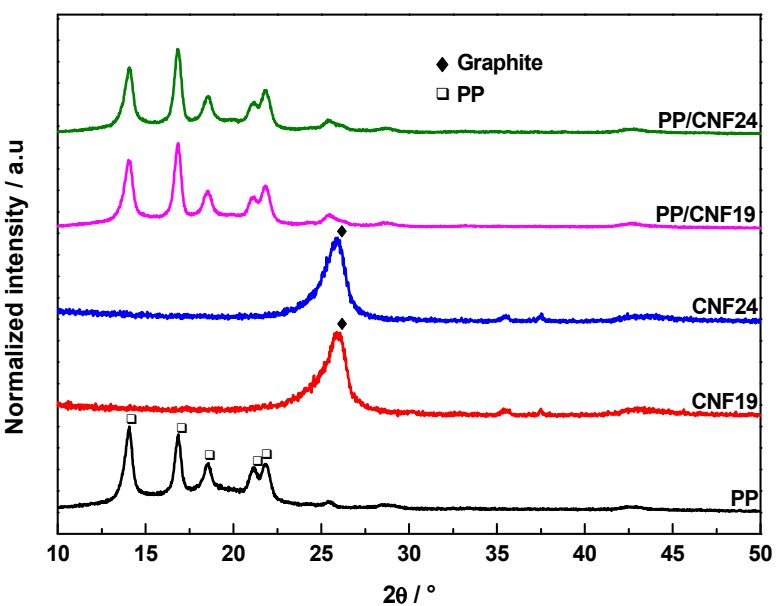

**Figure 3.** Representative XRD patterns of PP, as-received CNFs, and PP/CNF composites.

Figure 4 shows the Raman spectra obtained from PP, as-received CNFs and PP/CNF composites in the range between 600 and 1800 cm$^{-1}$. Polypropylene shows rich Raman spectra with modes in the range 800–1500 cm$^{-1}$ assigned to CH$_n$ stretching vibrations [25,26]. On their part, the as-received CNFs present the two bands expected in carbon nanostructures [15,21]. These are the D-band at ~1350 cm$^{-1}$, known as the disordered-induced mode [27], found when defects are present in the carbon aromatic structure, and the G-band around 1580 cm$^{-1}$, characteristic of the ideal graphitic lattice vibration [28]. As expected, the PP/CNF composites present the Raman signatures of the two materials. In particular, the most intense modes of PP (dotted lines in Figure 4) are clearly observable in PP/CNF composites. In the PP/CNF19 composite, the peak at 1460 cm$^{-1}$, corresponding to PP (numbered as 3 in Figure 4) shows the same intensity as do the G and D bands, whereas the peaks 1 and 2 of the PP hide the D-band of the CNFs [15]. These results suggest a stronger presence of PP in the analyzed area of that sample when compared to PP/CNF24 sample. Table 2 shows the peak position and the full with half maximum (FWHM) of the modes for CNFs and PP/CNF composites, determined by fitting the Raman spectra with Lorentzian functions. In addition, the in-plane graphitic domain size (L$_a$) in Table 2 is calculated according to $4.4/(I_D/I_G)$ [29] with I$_D$ and I$_G$ the intensities of the D and G bands, respectively. It is noteworthy that the FWHM$_G$ and FWHM$_D$ of PR 24 LHT XT are lower than those of PR 19 LHT XT, which suggests a higher degree of graphitization in the former [7]. Table 2 also shows that the G and D peak positions are practically the same for both CNFs and have slightly higher wavenumbers for both PP/CNF composites than for their CNF powders. Hence, the melt-mixing processing or the slight differences in interactions between the polypropylene and the CNF affect only to some extent the vibrational frequency of the native CNF bands. The intensity ratios between the D and G bands (I$_D$/I$_G$) are also calculated and presented in Table 2, since they are an important

parameter for quantifying the number of disordered (D), and ordered (G) carbon atoms [30]. Not surprisingly, the $I_D/I_G$ of PR 19 LHT XT is higher than in PR 24 LHT XT, again related to the higher degree of graphitization of the latter. Interestingly, the values of $L_a$ shown in Table 2 for all samples are similar to the larger crystal domain sizes observed by XRD for both CNFs (7 nm) and associated with the outer layers of the CNFs. In definitive, it is possible to conclude from the XRD and Raman analysis that the as-received CNFs are very similar, though PR 24 LHT XT grade seems to have a higher degree of graphitization according to the Raman analysis. In addition, the presence of PP in PP/CNF19 samples is more evident than in PP/CNF24 samples, which could be related with the higher number of CNFs at the studied content of 5 wt.% in PP/CNF composites produced with CNFs having the higher AR (PR 24 LHT XT).

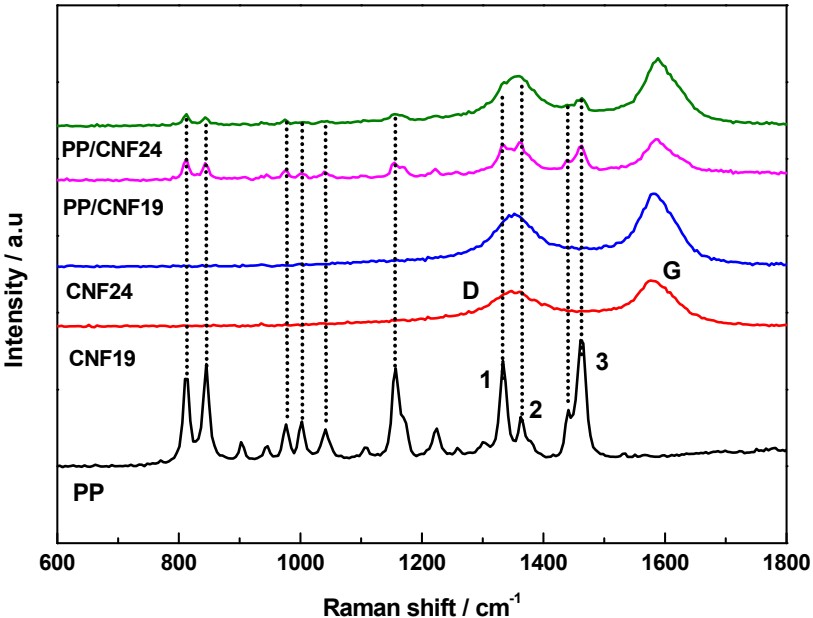

**Figure 4.** Raman spectra of PP, as-received CNFs and PP/CNF composites (data of PP, CNF 19 and PP/CNF19 taken with permission from ref. [15]. Data of CNF 24 taken with permission from ref. [21]).

**Table 2.** D and G peak positions (cm$^{-1}$) and respective full-width half maximum FWHM (cm$^{-1}$). Intensity ratio between D and G bands ($I_D/I_G$), obtained from the Raman fit, and $L_a$ (nm) in CNFs and PP/CNF composites. (Data of PP, CNF 19 and PP/CNF19 taken with permission from ref. [15]. Data of CNF 24 taken with permission from ref. [21]).

| Sample | $w_G$ (cm$^{-1}$) | FWHM$_G$ (cm$^{-1}$) | $w_D$ (cm$^{-1}$) | FWHM$_D$ (cm$^{-1}$) | $I_D/I_G$ | $L_a$ (nm) |
|---|---|---|---|---|---|---|
| CNF19 | 1580 | 90 | 1352 | 115 | 0.76 | 5.8 |
| CNF24 | 1583 | 85 | 1352 | 100 | 0.70 | 6.3 |
| PP/CNF19 | 1587 | 50 | 1353 | 75 | 0.70 | 6.3 |
| PP/CNF24 | 1587 | 65 | 1354 | 85 | 0.74 | 5.9 |

### 3.3. DSC Analysis

Figure 5 shows the DSC analysis for information about the influence of CNFs on PP's crystallization, which should affect the electrical properties of PP/CNF composites. In particular, the melting temperature ($T_m$) and degree of crystallinity ($\Delta X_c$) from the second heating scans are determined by:

$$\Delta X_C = \frac{\Delta H_M}{\Delta H_0 f_{PP}} \times 100\% \tag{1}$$

Here, $\Delta H_m$ is the melting enthalpy of the PP or PP part of the composites and $\Delta H_0 f_{PP}$ is the melting enthalpy of the 100% crystalline PP (207 J g$^{-1}$) [31]. Table 3 presents the corresponding values of $T_m$, $\Delta H_m$ and $\Delta X_c$. PP shows a melting peak at ~165 °C in accordance with other works [32], while PP/CNF19 and PP/CNF24 composites show lower melting temperatures of ~163 °C. This small decrease in $T_m$ is also in agreement with precedent works [33]. Interestingly, the results of Equation (1) exhibit a notable increase in $\Delta X_c$ from 34.5% corresponding to PP, to 51.1% and 55.1% for PP/CNF24 and PP/CNF19 composites, respectively, as a consequence of heterogeneous nucleation promoted by the CNFs into the PP [32,34].

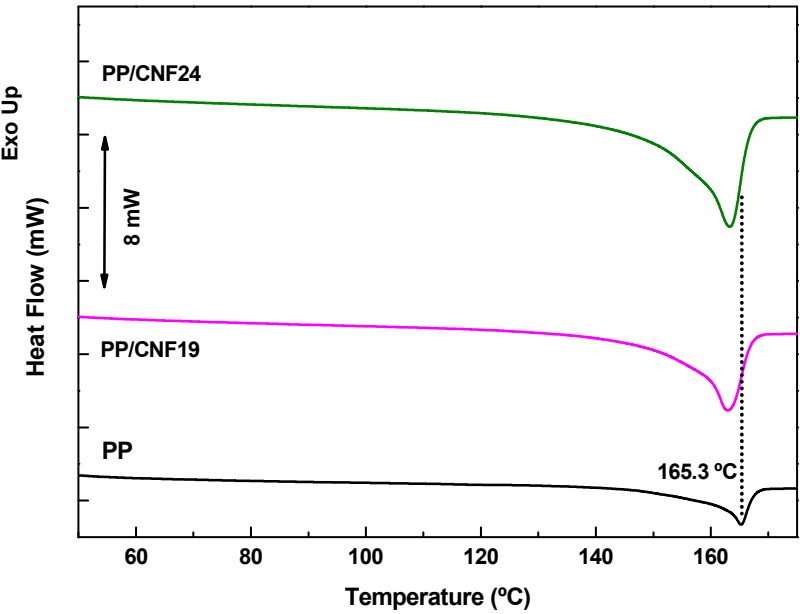

**Figure 5.** DSC thermographs of PP and PP/CNF composites, second heat.

**Table 3.** DSC data of neat PP and PP/CNF composites corresponding to the second heating scans.

| Sample | $T_m$ (°C) | $\Delta H_m$ (J g$^{-1}$) | $\Delta X_c$ (%) |
|--------|-----------|------------|------------|
| PP | 165.3 | 71.4 | 34.5 |
| PP/CNF19 | 162.9 | 107.7 | 55.1 |
| PP/CNF24 | 163.3 | 100.0 | 51.2 |

### 3.4. Thermoelectric Analysis of PP/CNF Composites at 40 °C

Figure 6 (squared symbols) and Table 4 represent the electrical conductivities of CNFs and PP/CNF composites at 40 °C. Interestingly, both CNFs present analogous values of ~132 S m$^{-1}$, equivalent to $8 \times 10^{-1}$ Ohm cm, higher than the $4 \times 10^{-3}$ Ohm cm reported for individual Pyrograf III CNFs [8]. As expected, the σ of PP/CNF composites is significantly lower than the σ of their corresponding CNFs. This change is attributed to the existence of PP chains wrapped around or being in near contact with the CNFs, which enhances the electrical contact resistance between adjacent CNFs [35]. Notably, despite the similar σ of CNFs (131 S m$^{-1}$), the PP/CNF24 composite shows a higher value of $62.7 \pm 7.0$ S m$^{-1}$, when compared with the PP/CNF19 composite ($16.5 \pm 0.7$ S m$^{-1}$). This is expected due to their higher AR of CNF24 meaning that at the constant content of 5 wt.%, a higher number (of thinner) CNFs are available to contribute to the electrically conductive network. Thus, this network is denser and stronger. Furthermore, as reported in [20], the better dispersion of CNF24 in PP contributes to more efficient networks with higher σ. Noticeably, the volume resistivity of the PP/CNF24 composite (1.3 Ohm cm) is in the range of PP composites melt-mixed with different commercial MWCNTs [1,36].

The bars in Figure 6 present the Seebeck coefficient at 40 °C of CNFs and PP/CNF composites. Similarly to their σ, the S of both CNFs is comparable, with values of ~−5.4 μVK$^{-1}$. Therefore, the CNFs show an n-type character, in contrast to most as-produced CNTs that are p-type due to their oxygen doping with the environment [37]. Accordingly, the S of PP/CNF composites is also negative (Table 4), and therefore, the CNFs give their n-type character to the PP/CNF composites. In particular, PP/CNF24 and PP/CNF19 present values of −4.4 ± 0.1 μVK$^{-1}$ and −3.8 ± 0.1 μVK$^{-1}$ at 40 °C, respectively. Hence, they have lower S-values than the CNFs in terms of absolute value. This variation of S leads to the possibility that PP may have an active role in the S-values of PP/CNF composites. To explain this, a recent study predicts a very slight electron donation from the outer layers of the CNFs to the surrounding PP molecular chains by the use of a semiempirical quantum chemical model [14]. In comparison with other reports, the negative Seebeck coefficients are lower than the values around −23 μV K$^{-1}$ reported in PP composites melt-mixed with 5 wt.% of nitrogen-doped MWCNTs [38]. More recently, promising S-values of −31.5 μV K$^{-1}$ have been found for melt-mixed PP/2 wt.% SWCNT composites with the addition of 5 wt.% polyvinylpyrrolidone (PVP), though their n-type character was lost after 6 to 18 months of storage under ambient conditions [39]. In addition, a higher value of −56.6 μVK$^{-1}$ was achieved in PP composites melt-extruded with 2 wt.% of single wall carbon nanotubes (SWCNTs) and 5 wt.% of copper oxide (CuO) after the addition of 10 wt.% of polyethylene glycol (PEG) during melt-extrusion [9].

Figure 6 (circular points) and Table 4 also present the power factor PF ($S^2$ σ) of CNFs and PP/CNF composites at 40 °C. The CNFs show a similar PF of $3.8 \times 10^{-3}$ μW m$^{-1}$ K$^{-2}$, associated with their comparable σ and S. Among the composites, which both have lower values than the CNFs, the PP/CNF24 composite achieved a higher PF of $1.2 \times 10^{-3}$ μW m$^{-1}$ K$^{-2}$ (compared to $2.4 \times 10^{-4}$ μW m$^{-1}$ K$^{-2}$ of PP/CNF19). Notably, this PF is higher than that of some melt-mixed PP composites with 5 wt.% of nitrogen-doped MWCNTs above mentioned [38].

The figure of merit zT at 40 °C of PP/CNF composites is also shown in Table 4 and Figure 6 (triangle symbols) after obtaining experimentally their thermal conductivity as described in Section 2.5. Interestingly, the PP/CNF24 composite shows higher thermal conductivity at 40 °C (0.29 W m$^{-1}$ K$^{-1}$ compared to 0.25 W m$^{-1}$ K$^{-1}$ for PP/CNF19), which correlates with better CNF24 dispersion in the PP [20]. This is different from earlier findings reported on the same composites but compression molded under different conditions and measured in a different laboratory, where slightly higher thermal conductivities were found for PP/CNF19 composites compared to PP/CNF24 composites [40]. The combination of higher σ and a more negative S value, despite the higher thermal conductivities, results in PP/CNF24 in a higher zT of $1.3 \times 10^{-6}$ (compared to $3.0 \times 10^{-7}$ for PP/CNF19). It must be noticed that higher zT values (3.0 to $3.3 \times 10^{-5}$ at 40 °C) were already reported for PP composites filled with 2 wt.% of SWCNTs, and PP filled with 2 wt.% of branched MWCNTs [23].

**Table 4.** Electrical conductivity σ, Seebeck coefficient S, power factor PF, thermal conductivity k, and figure of merit zT of CNFs and PP/CNF composites at 40 °C. (Data for CNF19 and PP/CNF19 taken with permission from ref. [15]. Data of CNF 24 taken with permission from ref. [21]).

| Sample | σ (S m$^{-1}$) | S (μV K$^{-1}$) | PF (μW m$^{-1}$K$^{-2}$) | k (W m$^{-1}$K$^{-1}$) | zT |
|---|---|---|---|---|---|
| CNF19 | 131.5 ± 19 | −5.4 ± 0.2 | $3.8 \times 10^{-3}$ | – | – |
| CNF24 | 131.6 ± 0.1 | −5.4 ± 0.1 | $3.8 \times 10^{-3}$ | – | – |
| PP/CNF19 | 16.5 ± 0.7 | −3.8 ± 0.1 | $2.4 \times 10^{-4}$ | 0.25 | $3.0 \times 10^{-7}$ |
| PP/CNF24 | 62.7 ± 7 | −4.4 ± 0.1 | $1.2 \times 10^{-3}$ | 0.29 | $1.3 \times 10^{-6}$ |

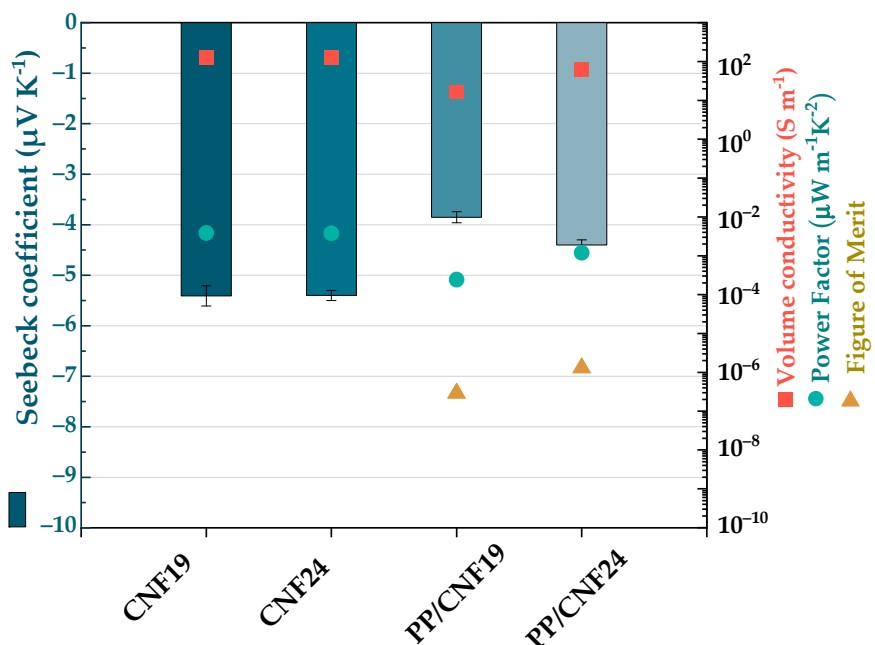

**Figure 6.** Electrical conductivity (squared symbols), Seebeck coefficient (bars), power factor (circle symbols), and figure of merit (triangle symbols) at 40 °C of CNFs and PP/CNF composites. (Data for CNF19 and PP/CNF19 taken with permission from ref. [15]. Data of CNF 24 taken with permission from ref. [21]).

### 3.5. Thermoelectric Analysis of PP/CNF Composites from 40 °C to 100 °C

The thermoelectric properties σ (T) and S (T) from 40 °C to 100 °C of the CNFs and PP/CNF composites are represented in Figure 7 and Table S3 of supporting information. As noted in the previous section, electrical conductivities of ~131.5 S m$^{-1}$ and ~131.6 S m$^{-1}$ at 40 °C are obtained for carbon nanofibers PR 19 LHT XT and PR 24 LHT XT, respectively. At 100 °C, the σ of both decreases up to ~127 S m$^{-1}$ and ~123.9 S m$^{-1}$, respectively. Thereby, both CNFs show a positive temperature effect (dρ/dT > 0). In particular, the CNFs PR 24 LHT XT suffer a stronger decrease in σ in this interval of temperature than PR 19 LHT XT grade. Likewise, the PP/CNF19 composite shows a positive temperature effect, where σ decreases gradually from ~16.5 S m$^{-1}$ at 40 °C to ~13.9 S m$^{-1}$ at 100 °C. However, the σ of PP/CNF24 composites is more intricate. First, σ decreases from ~62.7 S m$^{-1}$ at 40 °C to ~56.0 S m$^{-1}$ at 60 °C, then increases slightly to ~56.8 S m$^{-1}$ at 80 °C, and finally it drops to ~53.2 S m$^{-1}$ at 100 °C. Figure 7 (as red circle icons) and Table S3 also introduce the S (T) of the CNFs. It is clear that the n-type character of the CNFs at 40 °C remains negative at the rest of the temperatures. In particular, PR 19 LHT XT and PR 24 LHT XT present S-values of −5.4 ± 0.2 μV K$^{-1}$ and −5.4 ± 0.1 μV K$^{-1}$ at 40 °C, which rises gradually (in absolute value) up to −5.8 ± 0.1 μV K$^{-1}$ and −5.9 ± 0.1 μV K$^{-1}$ at 100° C (Table S3), respectively. The S (T) of PP/CNF19 and PP/CNF24 composites (red triangle symbols in Figure 7) show S-values from −3.8 ± 0.1 μV K$^{-1}$ and −4.4 ± 0.1 μV K$^{-1}$ at 40 °C to −4.3 ± 0.1 μV K$^{-1}$ and −6.1 ± 0.1 μV K$^{-1}$ at 100 °C, respectively (Table S3). Thus, the S (T) of the PP/CNF19 composite shows negative S-values gradually increasing (in absolute value) within this range of temperatures, while the effect of temperature in the increasing of S is more pronounced for the PP/CNF24 composites. Ultimately, it can be concluded that despite both Pyrograf® III grades (PR 19 LHT XT and PR 24 LHT XT) presenting similar σ (T) and S (T) in the interval of 40 °C–100 °C (Figure 7a), the σ (T) and S (T) of their derivative PP/CNF19 and PP/CNF24 composites are not completely analogous (Figure 7b).

The power factor PF as a function of the temperature of the CNFs and PP/CNF composite is shown in Table S3. At 40 °C, both CNFs present a comparable PF of ~3.8 × 10$^{-3}$ μW m$^{-1}$ K$^{-2}$, which increases up to ~4.3 × 10$^{-3}$ μW m$^{-1}$ K$^{-2}$ at 100 °C. While for PP/CNF samples, the PP/CNF24 composite show the highest PF of ~1.2 × 10$^{-3}$ μW m$^{-1}$ K$^{-2}$

at 40 °C, which enhances up to ~$2 \times 10^{-3}$ µW m$^{-1}$ K$^{-2}$ at 100 °C. This corresponds to a zT of ~$1.3 \times 10^{-6}$ at 40 °C for PP/CNF24, which increases up to ~$2.5 \times 10^{-6}$ at 100 °C. This means that the PP/CNF24 composite presents higher PF and zT values than the PP/CNF19 composite for this range of temperatures, despite both CNFs (Pyrograf® III PR 24 LHT XT PR 19 LHT XT) exhibiting similar σ (T) and S (T) as illustrated in this section. Hence, the combination of the higher aspect ratio and better dispersion of CNFs PR 24 LHT XT within the PP [20] could be the principal reasons behind the better thermoelectric performance of PP/CNF24 composites.

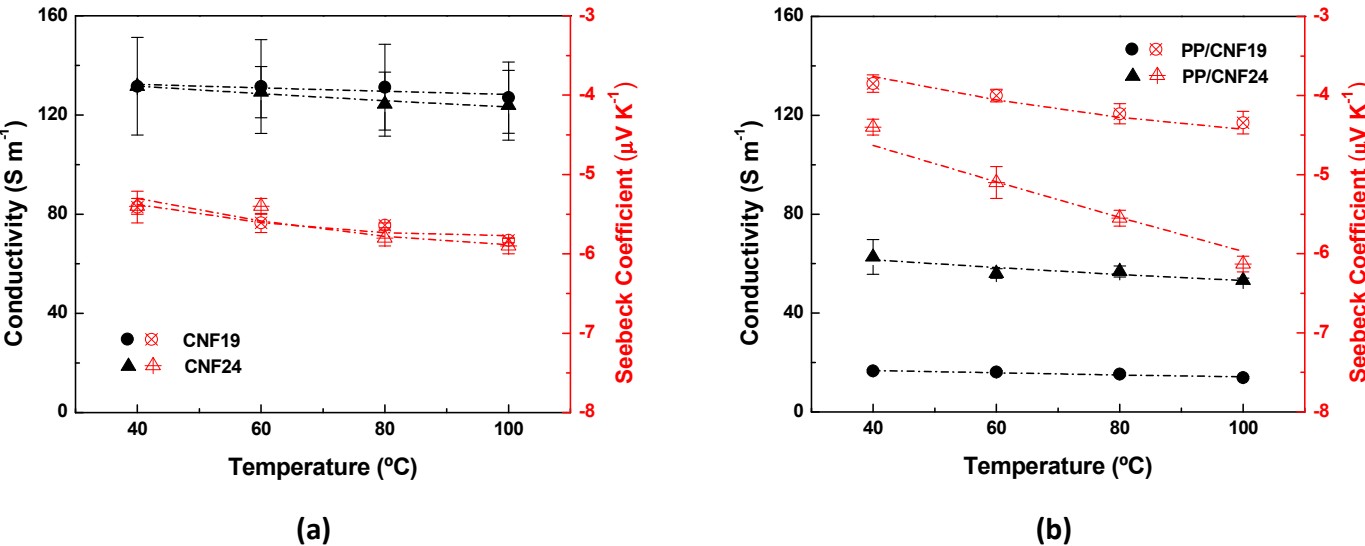

(a)                                            (b)

**Figure 7.** Electrical volume conductivity (black symbols) and Seebeck coefficient (red symbols): (**a**) σ and S of CNFs, (**b**) σ and S of PP/CNF composites. The black and red dash lines represent the fitting with Equations (2) and (3), respectively. (Data for CNF19 and PP/CNF19 taken with permission from ref. [15]. Data of CNF 24 taken with permission from ref. [21]).

*3.6. Electrical Volume Conductivity and Seebeck Coefficient Modelling of CNFs and PP/CNF Composites*

The 3D variable-range hopping (VRH) model is applied to examine the σ (T) of the CNFs powder and PP/CNF composites in the temperature of 40 °C to 100 °C [16]:

$$\sigma(T) = \sigma_0 \exp\left[\pm\left(\frac{T_C}{T}\right)^{\frac{1}{4}}\right] \tag{2}$$

Here, $\sigma_0$ is the conductivity at an infinite temperature, $T_C \equiv \frac{|W_D|}{k_B}$ is a characteristic temperature defined by the average energy potential barrier ($W_D < 0$) or potential well ($W_D > 0$), respectively, and $k_B$ is the Boltzmann's constant. When $W_D > 0$, Equation (2) depicts a thermally activated hopping mechanism across a random network of potential wells, while when $W_D < 0$, it defines a thermally activated scattering across a random distribution of impurities or structural defects. More specifically, the values of $\sigma_0 = 63.7$ S m$^{-1}$, $T_C = 89.7$ K, and $W_D = -7.7$ meV as presented in Table 5 are obtained from Equation (2) for PR 19 LHT XT CNFs, while PR 24 LHT XT CNFs show values of $\sigma_0 = 28.3$ S m$^{-1}$, $T_C = 1.7 \times 10^3$ K, and $W_D = -150$ meV. Interestingly, the $T_C$ of PR 24 LHT XT is one order higher than that of some SWCNT mats ($2.5 \times 10^2$ K) [41], likewise, its $W_D$ in absolute value is one order higher than the activation energy calculated for n-type graphitized carbon fibers in the 250–750 K interval (60 meV) [42]. Noticeably, the CNFs show both negative $W_D$. This sign can result from the presence of defects or elements different from carbon, such as the oxygen observed by X-ray photoelectron spectroscopy (XPS) for both CNFs [15,43]. More precisely, these defects could activate a thermal-enhanced backscattering mechanism due to the presence of virtual bound-states, represented as sharp peaks close to the $E_F$ of their density of states [44].

The σ (T) of PP/CNF composites was also studied with the 3D VRH model, from which PP/CNF19 composites present values of $\sigma_0 = 0.3$ S m$^{-1}$, $T_C = 73.6 \times 10^3$ K, and $W_D = -6.3$ eV, while PP/CNF24 composites exhibit values of $\sigma_0 = 1.9$ S m$^{-1}$, $T_C = 45.4 \times 10^3$ K, and $W_D = -3.9$ eV. Consequently, the CNFs and their PP/CNF composites present all negative $W_D$, and therefore, their σ (T) is described by a thermally activated backscattering mechanism.

**Table 5.** Constants $\sigma_0$, $T_C$, and $W_D$ determined from VRH model of CNFs and PP/CNF composites.

| Sample | $\sigma_0$ (S m$^{-1}$) | $T_C$ (K) | $W_D$ (eV) |
|---|---|---|---|
| CNF19 | 63.7 | 89.7 | $-7.7 \times 10^{-3}$ |
| PP/CNF19 | 0.3 | $73.6 \times 10^3$ | $-6.3$ |
| CNF24 | 28.3 | $1.7 \times 10^3$ | $-0.15$ |
| PP/CNF24 | 1.9 | $45.4 \times 10^3$ | $-3.9$ |

The S (T) of CNFs and PP/CNF composites is depicted by the theoretical model proposed for describing the nonlinear behavior of doped MWCNT mats [17]:

$$S(T) = bT + \frac{cT_p}{T^2} \frac{\exp\left(\frac{T_p}{T}\right)}{\left[\exp\left(\frac{T_p}{T}\right) + 1\right]^2} \tag{3}$$

Here, bT represents the metallic (linear) term, c is a constant, and $T_p = (E_P - E_F)/k_B$ where $k_B$ is Boltzmann's constant, $E_F$ is the Fermi energy level, and $E_P$ is the energy corresponding to the sharply varying and localized states near $E_F$ in the density of states due to the contribution of impurities [44]. The best fit of S (T) with Equation (3) for PR 19 LHT XT (Table 6) shows that the first term b is positive with $6.1 \times 10^{-3}$ μV K$^{-2}$, while the second term c is negative with $-1.8 \times 10^4$ μV and $T_p = 981.3$ K, and a $E_P - E_F = 0.085$ eV. In the case of CNFs PR 24 LHT XT, the term b is $4.2 \times 10^{-3}$ μV K$^{-2}$, while c is $-1.8 \times 10^4$ μV and $T_p = 1014.3$ K, yielding a $E_P - E_F = 0.087$ eV. From these results, it is deduced that both Pyrograf® III grades (PR 19 LHT XT and PR 24 LHT XT) practically share the same S (T) in the interval of 40 °C–100 °C. Furthermore, the negative sign of c found in both CNFs can be interpreted as the resonances near the $E_F$ at the density of states caused by impurities or defects present in the CNF structure [17]. Likewise, the S (T) of PP/CNF composites is also fitted by Equation (3). The best fit for PP/CNF19 composites yields values of b = $4.7 \times 10^{-3}$ μV K$^{-2}$, c = $-1.6 \times 10^4$ μV, $T_p = 1092.7$ K, and $E_P - E_F = 0.094$ eV. Thereby, the fittings obtained by Equation (3) for PP/CNF19 are very similar to the parameters calculated for the CNFs PR 19 LHT XT used in their preparation (Table 6). However, that is not the case with the modeling of PP/CNF24 composites, which yields values of b = $-9.1 \times 10^{-3}$ μV K$^{-2}$, c = $-1.2 \times 10^4$ μV, $T_p = 1446.5$ K, and $E_P - E_F = 0.125$ eV. In particular, it is remarkable the negative sign found for the term b, in contrast to the positive b of the CNFs PR 24 LHT XT applied in their melt-production. To understand this fact, it is necessary to realize that the two summands in Equation (3) describe two mechanisms occurring in parallel. One is defined by the first summand bT (metallic term), representing the contribution from nearly free charge carriers, which can be positive charge carriers (holes) if b > 0, or negative charge carriers (electrons) if b < 0. Therefore, the negative sign of b found in PP/CNF24 composites means that n-type doping from the PP may be inferred in PP/CNF24 composites. This is an unexpected result, since as it was discussed in Section 3.4, a slight electron withdrawing from the external layers of CNFs by the PP matrix would be foreseen as in the case of PP/CNF19 composites [14]. In short, it can be deduced from the σ (T) and S (T) modeling that the σ (T) of CNFs and PP/CNFs is described by a thermally activated backscattering mechanism. Nevertheless, the S (T) of PP/CNF24 composites, unlike the S (T) of PP/CNF19 composites, does not follow the S (T) depicted by their corresponding CNFs (PR 24 LHT XT).

**Table 6.** Constants b, c, $T_P$ and $E_P - E_F$ obtained by Equation (3) of CNFs and PP/CNF composites.

| Sample | b ($\mu VK^{-2}$) | c ($\mu V$) | $T_p$ (K) | $E_p - E_F$ (eV) |
|---|---|---|---|---|
| CNF19 | $6.1 \times 10^{-3}$ | $-1.8 \times 10^4$ | $9.8 \times 10^2$ | $8.5 \times 10^{-2}$ |
| PP/CNF19 | $4.7 \times 10^{-3}$ | $-1.6 \times 10^4$ | $1.1 \times 10^3$ | $9.4 \times 10^{-2}$ |
| CNF24 | $4.2 \times 10^{-3}$ | $-1.8 \times 10^4$ | $1.0 \times 10^3$ | $8.7 \times 10^{-2}$ |
| PP/CNF24 | $-9.1 \times 10^{-3}$ | $-1.2 \times 10^4$ | $1.4 \times 10^3$ | $1.2 \times 10^{-1}$ |

## 4. Conclusions

The electrical conductivity (σ) and Seebeck coefficient (S) of two industrial-grade carbon nanofibers (CNFs) and their melt-extruded polypropylene (PP) composites with 5 wt.% of CNFs in the temperature range between 40 °C and 100 °C are compared in this study. The experimental analysis together with the modeling of σ (T) and S (T) reveals that the filler aspect ratio (AR) and dispersion affect the thermoelectrical (TE) properties of the resulting PP/CNF composites. In particular, the PP/CNF composite produced with CNFs with higher AR and higher levels of graphitization (Pyrograf® III PR 24 LHT XT), achieves higher values of σ at the same content of CNFs (5 wt.%). These findings lead to the PP/CNF24 composites having a higher PF and figure of merit (zT) and thus better TE performance than PP/CNF19 composites. From the σ(T) fitting with the 3D variable-range hopping (VRH) model, it is deduced that the CNFs clearly drive the σ(T) of their corresponding PP/CNF composites. However, quite unexpectedly, this is not the case for the S(T) of the PP/CNF24 composites, which after modeling with the same theoretical model proposed for the S(T) of PP/CNF19 composites, points to n-type doping from the PP to the CNFs activated with the temperature.

**Supplementary Materials:** The following supporting information can be downloaded at: https://www.mdpi.com/article/10.3390/jcs7040173/s1 [45].

**Author Contributions:** All authors of this manuscript contributed to the development of this work. A.J.P. conceived the study and contributed to formal analysis, data curation, and writing-the original draft preparation. P.P. and B.K. were responsible for thermoelectric analysis and review writing. E.M. was responsible for thermoelectric modeling and reviewing. M.F.C. was responsible for the Raman analysis and its discussion. C.J.T. and A.R.M. were responsible for the XRD analysis and discussion. All authors have read and agreed to the published version of the manuscript.

**Funding:** A.J.P. gratefully acknowledges support from FCT-Foundation for Science and Technology by the project UID/CTM/00264/2021 of 2C2T under the COMPETE and FCT/MCTES (PIDDAC) co-financed by FEDER through the PT2020 program and "plurianual" 2020–2023 Project UIDB/00264/2020. C.J.T. acknowledges the funding from FCT/PIDDAC through the Strategic Funds project reference UIDB/04650/2020-2023. E.M. acknowledges support from Project ANID PIA Anillo ACT/192023.

**Data Availability Statement:** The data presented in this study are available on request from the corresponding author.

**Acknowledgments:** The authors would like to thank Delfim Soares from CMEMS of the University of Minho for the access to the DSC equipment and DSC data.

**Conflicts of Interest:** The authors declare no conflict of interest.

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
