# Peer review of "Comparative Thermoelectric Properties of Polypropylene Composites Melt-Processed Using Pyrograf® III Carbon Nanofibers"

_jcs, doi:10.3390/jcs7040173_

Round 1
Reviewer 1 Report
After read carefully this present manuscript, I feeled confused to accept this present manuscript under its current form, for the publication in your journal Composites Sciences. In many cases the authors compared their most recent results to the previous one (ref 15). Unfortunately, I have the strong feeling they had divided their results to make a second article. By the way, if the author could add further analysis such as XRD, viscosity measurement and mechanical testing, their work should appear more complete and afford new extent to their researches. (for example).
I hope the authors could at least evaluate the thermoelectric properties under deformation and also test effects of defects or ruptures on the intrinsec thermoelectric properties of their composites materials.
Reviewer 2 Report
The authors studied the thermoelectric properties of polypropylene composites melt-processed using Pyrograf® III carbon nanofibers with different aspect ratio and enhanced the understanding of the role of the filler aspect ratio on the thermoelectrical properties of the PP/CNF composites. Overall, the results are interesting and the manuscript is well-organized and written. I therefore recommend the publication of this manuscript in Journal of Composites Science after minor revision. However, there are still two issues that require a response from the author.
1) It is recommended that the authors describe the purpose and significance of the study in the abstract.
2) The paper lacks analysis and discussion on the thermoelectric mechanism, such as which of the several factors, aspect ratio, distribution, dispersion and orientation, will have a greater effect on the thermoelectric performance under what conditions? What other geometrical or surface features of carbon nanofibers besides aspect ratio affect the thermoelectric properties of the composite system? It is suggested to add relevant discussions.
Reviewer 3 Report
1. It is recommended to add photos of manufactured samples (rods, peletz, plates) in section 2.1. Materials and their Processing.
2. It is recommended to add a photo of the equipment for measuring seebeck coefficient and electrical volume resistivity in section 2.5. Thermoelectric analysis.
3. It is recommended to discuss how the results obtained can be applied in the engineering practice of designing products made of thermoplastic composites. In which products can PP/CNF composites be used with the obtained thermoelectric properties level?
4. How are you planning to continue this research? For example, conducting mechanical tests of the obtained PP/CNF composites.
Round 2
Reviewer 1 Report
After checked carefully the actual version of the manuscript, I think this article could be accepted for publication in your journal J. Compos Sci under this form.